# The Associations between Immunological Reactivity to the Haptenation of Unconjugated Bisphenol A to Albumin and Protein Disulfide Isomerase with Alpha-Synuclein Antibodies

**DOI:** 10.3390/toxics7020026

**Published:** 2019-05-06

**Authors:** Datis Kharrazian, Martha Herbert, Aristo Vojdani

**Affiliations:** 1Harvard Medical School, 25 Shattuck St, Boston, MA 02115, USA; martha.herbert@mgh.harvard.edu; 2TRANSCEND Research Laboratory, Department of Neurology, Massachusetts General Hospital, 149 13th Street, Boston, MA 02129, USA; 3Department of Preventive Medicine, Loma Linda University School of Medicine, 24785 Stewart Street, Loma Linda, CA 92354, USA; drari@msn.com; 4Immunosciences Lab., Inc., 822 S. Robertson Boulevard, Suite 312, Los Angeles, CA 90035, USA

**Keywords:** bisphenol A, protein disulfide isomerase, alpha-synuclein antibodies, Parkinson’s disease, neurotoxicity

## Abstract

Patients with Parkinson’s disease (PD) have increased susceptibility to bisphenol A (BPA) exposure since they have an impaired biotransformation capacity to metabolize BPA. PD subjects have reduced levels of conjugated BPA compared to controls. Reduced ability to conjugate BPA provides increased opportunity for unconjugated BPA to bind to albumin in human serum and protein disulfide isomerase on neurons. Once unconjugated BPA binds to proteins, it changes the allosteric structure of the newly configured protein leading to protein misfolding and the ability of the newly configured protein to act as a neoantigen. Once this neoantigen is formed, the immune system produces antibodies against it. The goal of our research was to investigate associations between unconjugated BPA bound to human serum albumin (BPA–HSA) antibodies and alpha-synuclein antibodies and between Protein Disulfide Isomerase (PDI) antibodies and alpha-synuclein antibodies. Enzyme–linked immunosorbent assay was used to determine the occurrences of alpha-synuclein antibodies, antibodies to BPA–HSA adducts, and PDI antibodies in the sera of blood donors. Subjects that exhibited high levels of unconjugated BPA–HSA antibodies or PDI antibodies had correlations and substantial risk for also exhibiting high levels of alpha-synuclein antibodies (*p* < 0.0001). We conclude that there are significant associations and risks between antibodies to BPA–HSA adducts and PDI antibodies for developing alpha-synuclein antibodies.

## 1. Introduction

Parkinson’s disease (PD) is the second most common neurodegenerative disease, and 95% of cases are sporadic and unassociated with familial PD [1]. Therefore, environmental risk factors leading to PD is a vital area of investigation. The specific mechanisms of how environmental chemicals promote the development of PD are an area of ongoing research. Epidemiological, animal, and human research suggest that the development of alpha-synuclein aggregates is a key feature of PD [1].

The mechanisms of how environmental chemicals lead to the development of alpha-synuclein aggregates are particularly important for ubiquitous environmental compounds. Bisphenol A (2,2-bis(4-hydroxyphenyl) propane; BPA) is one of the most common chemicals to which humans are exposed to because BPA is the key chemical used to manufacture plastic products, such as water bottles, utensils, bags, and containers [2]. Frequent exposure to products containing BPA has led to elevated levels of BPA in a large percentage of human samples in the United States population [3,4]. On an international scale, human urinary and tissue biomonitoring studies also confirm increased levels of BPA in humans worldwide [5]. The extensive use of BPA throughout the world has made exposure to BPA an inescapable feature of modern society.

Patients with PD have increased risk and susceptibility to BPA exposure from various mechanisms [6,7,8,9,10]. BPA is cleared from the body through hepatic conjugation with glucuronic acid, and this biotransformation pathway is compromised in patients suffering from PD [6]. Landolfi et al. [6] reported that PD subjects were found to have statistically significant reduced levels of conjugated BPA compared to controls, and this variant in BPA metabolism may be associated with the increased risks seen in PD subjects. The impaired ability to clear BPA can potentiate its metabolic effects and the potential for unconjugated BPA to bind to tissue proteins [7,8,9,10].

When chemicals such as BPA enter the body, they can induce immune reactivity by binding to proteins in a process called haptenation [11]. Patients with PD have increased susceptibility to BPA exposure since they have impaired biotransformation capacity to metabolize BPA. Reduced ability to conjugate BPA provides increased opportunity for unconjugated BPA to bind to albumin in human serum. Once unconjugated BPA binds to albumin and changes the allosteric structure of the protein (BPA–HSA), it then can act as a neoantigen and upregulate the immune response. Elevated levels unconjugated BPA–HSA antibodies have been reported in subjects with neurological inflammatory reactions and are also highly correlated with PDI antibodies [12]. In addition to albumin, BPA can bind directly to PDI found on neurons [13]. The binding of BPA to PDI can change the allosteric function of PDI and promote protein misfolding and the production of antibodies to the protein [13,14,15]. These mechanisms may play a role in the neurodegenerative process and are illustrated in Figure 1.

The goal of our research was to investigate associations between unconjugated BPA bound to human serum albumin (BPA–HSA) antibodies and alpha-synuclein antibodies and between PDI antibodies and alpha-synuclein antibodies. Antibodies formed against alpha-synuclein are an early serum biomarker to identify the development of alpha-synuclein aggregation, which is the hallmark of PD [16,17].

## 2. Materials and Methods

### 2.1. Blood Samples

This research received IRB approval from Partner’s Human Research Committee at Massachusetts General Hospital (Protocol #2106P002738/MGH; date of approval: 20 December 2016). Sera from 94 random human donors aged 18 to 65 years were purchased from Innovative Research Inc. (Southfield, MI, USA). Each sample of blood was tested according to the guidelines of the United States Food and Drug Administration to identify hepatitis C, RNA, hepatitis B surface antigen, antibodies to HIV, HIV-1 RNA, and syphilis. The results were negative for all of the samples used in our study. There was no assessment for cardiovascular, metabolic, or autoimmune diseases, and we did not have any further medical information of the subjects we used in our samples. Tests samples were selected and arranged to reflect the adult population of the United States by sex and race. Adult subjects were separated by gender and then by racial distribution to reflect the U.S. population. Serum samples included: 60% Caucasians; 20% Hispanics; and 20% African American and other races.

### 2.2. Proteins and Chemicals

BPA, HSA, bovine serum albumin (BSA), and alpha-synuclein were all purchased from Sigma Aldrich (St. Louis, MO, USA), and PDI was acquired from BioSynthesis (Lewisville, TX, USA) to develop antigen-coated ELISA plates.

### 2.3. Preparation of BPA bound HSA

Preparation of unconjugated BPA valeric acid bound to HSA was performed by dissolving one gram of HSA in 100 mL of 0.01 M phosphate buffered saline (PBS) pH 7.4. A total of 100 mg of BPA was dissolved in 10 mL of 0.01 M PBS pH 7.4. to each tube and the chemicals were separately added to the protein mixture in a dropwise fashion. Then 100 mg of N-hydroxysulfosuccinimide sodium salt in 10 mL of distilled water was prepared in a separate tube and then added to the mixture in a dropwise fashion. The mixtures were kept for 1 h at room temperature and then 4 h at 4 °C. Dialysis was used to remove the nonreacted small molecules using a molecular cutoff of 8,000 Da. Sodium dodecyl sulfate (SDS) gel electrophoresis and a shift in band configuration were used to confirm conjugation of BPA binding to HSA (BPA–HSA). Additionally, we undertook a spectrographic analysis of the conjugate until there was an increase in absorption from 230 to 260 nm in order to confirm that BPA was covalently linked to HSA.

### 2.4. Measurement of Antibody by Enzyme-Linked Immunosorbent Assay (ELISA)

BPA (unconjugated), PDI, and alpha-synuclein were dissolved in PBS at a concentration of 1.0 mg/mL. The solution was then diluted to 1:100 in 0.1 M carbonate-bicarbonate buffer, pH 9.5, and 100 μL. The mixture was then added to each well of a polystyrene flat bottom ELISA plate. The plates were washed three times with 200 μL Tris-buffered saline (TBS) containing 0.05% Tween 20, pH 7.4, after being incubated overnight at 4 °C. The nonspecific binding of immunoglobulins was prevented by adding a mixture of 2% BSA in TBS. The solution was then incubated overnight at 4 °C. The plates were washed as described previously. Serum samples were added to duplicate wells after being diluted to 1:100 in 0.1 M PBS Tween containing 1% BSA and incubated at room temperature for one hour. Sera with high levels of antibodies from patients with neuroimmune disorders were used as positive controls. The plates were washed again. Then we added alkaline phosphatase goat antihuman IgA, IgG, or IgM F(ab′)2 fragments (KPI, Gaithersburg, MD, USA) at an optimal dilution of 1:400–1:2000 in 1% HSA–TBS were added to different wells. The plates were then incubated for an additional 1 h at room temperature. The plates were washed five times with TBS Tween buffer. The enzyme reaction was initiated by adding 100 μL of paranitrophenylphosphate (PNPP) in 0.1 mL diethanolamine buffer 1 mg/mL containing one mM MgCl2 and sodium azide at a pH of 9.8. The reaction was terminated 45 min later with 50 μL of 1N NaOH. The optical density (OD) was registered at 405 nm through a microtiter reader. To detect nonspecific binding, we used several control wells containing all reagents except human serum. Furthermore, other wells were coated with different antigens. All the other reagents were added and the ODs were recorded.

Additionally, in each assay, we included sera of patients with Parkinson’s disease with moderate to high levels of alpha-synuclein, PDI, and BPA antibodies. An optimal dilution of the sera at 1:200–1:400 and incubation for 45 min resulted in optical densities of 1.1 ± 5%, from which we chose the three best to use as a calibrator in each assay, one for BPA–HSA, one for PDI, and one for alpha-synuclein. Then, the ODs were converted to ELISA index using the following formula:$$\mathrm{ELISA}\;\mathrm{Index}=\frac{\mathrm{OD}\;\mathrm{of}\;\mathrm{sample}-\mathrm{OD}\;\mathrm{of}\;\mathrm{blank}}{\mathrm{OD}\;\mathrm{of}\;\mathrm{calibrator}-\mathrm{OD}\;\mathrm{of}\;\mathrm{blank}}$$

Because the OD of our calibrators was 1.1 and the blank wells were about 0.1 ± 10%, the calculations of our indices were 1.0 ± 5%, which were very close to ODs, and therefore we decided to report the original ODs in our analyses.

### 2.5. Statistical Analysis

Statistical analyses were performed to study the relationships between BPA, PDI, and alpha-synuclein antibodies using Pearson’s correlation coefficient, two-sample *t*-test, and the chi-squared test. A Bonferroni adjustment was conducted for multiple comparisons. A separate analysis was conducted for each of three immunoglobulins: IgG, IgA, and IgM. The magnitude of the correlative relationships is reported. STATA software package 14.2 was used to conduct all analyses. We did not have adequate power to stratify our samples for gender, sex, or race in our study. Our analyses were only powered to analyze groups above and below the mean.

## 3. Results

The optical density (OD) outcomes of ELISA results were measured at 405 nm. The OD measurements for IgG, IgM, and IgA were measured for both BPA–HSA and PDI. The results are presented in Figure 2. Between 62% and 67% of the OD measurements were below the mean for each immunoglobulin. There was a significant difference in the optical density means for those subjects that were above the mean OD compared to subjects that were below the mean OD in all tested groups (*p*-value < 0.0001).

Risk ratio (RR) calculations were conducted in two groups that included (1) subjects with antibodies above the mean, and (2) subjects with antibodies below the mean, for BPA–HSA antibodies and PDI antibodies to determine if there were any risk for exhibiting elevated alpha-synuclein antibodies. There was no identified risk for exhibiting elevated alpha-synuclein antibodies with subjects that had their BPA–HSA antibodies or PDI antibody levels below the mean. However, there was a significant risk (RR between 3.4 and 16.7) of exhibiting alpha-synuclein antibodies if the subjects had elevated (above the mean) BPA–HSA antibodies or PDI antibodies. These risk ratios are listed in Table 1.

There were also strong correlations and linear relationships (*r*-values between 0.5 and 0.8) for BPA–HSA and PDI antibodies to alpha-synuclein antibodies if the OD levels were above the mean, as shown in Figure 3. There were weak to no significant correlations with alpha-synuclein antibodies if BPA–HSA or PDI antibody levels were below the mean. Of the samples tested, approximately 30% of the samples had OD measurements that were above the mean, and it is only in these subjects that demonstrate correlation and risk of alpha-synuclein antibodies.

## 4. Discussion

Exposure to BPA leads to distribution of the chemical throughout the body. Once the chemical enters the systemic circulation of the body, it can be detected in serum samples [5]. BPA is then converted into a water-soluble metabolite by phase I and phase II hepatic biotransformation pathways, after which it is excreted in the urine [6]. Chemicals such as BPA also have the potential to bind to various proteins in the body, such as albumin and PDI, while they are in circulation [13,14]. When a chemical binds to a protein, it changes its allosteric structure, and the newly configured protein can misfold and also act as new antigens to the immune system [11]. When this occurs, the humoral system produces antibodies against these chemical-bound neoantigens [12,15]. A novel feature of our study was to investigate if neoantigens formed by BPA from haptenation may be associated with mechanisms that occur in synucleinopathies.

Our study found considerable relationships between antibodies formed against BPA-bound-to-albumin antibodies and antibodies formed against alpha-synuclein. An important feature of our study was the identification of antibodies formed against unconjugated BPA-bound-to-protein (BPA–HSA) and not merely the detection of unconjugated BPA levels. In our study, we focused on reactivity to neoantigens formed by BPA bound to proteins as a unique immunological mechanism.

We did not measure concentrations of BPA in our study since elevated BPA levels in biofluids is a ubiquitous finding. The perpetual exposure of BPA to humans is an established feature of modern society due to the extensive use of plastic products. Detectible levels of BPA in human serum and urine samples are identified in the majority of the population. More than 90% of the samples tested in the United States (U.S.) population have measurable BPA levels in biofluids (4). However, antibodies produced against BPA-bound-to-protein are an independent feature of only a subset of the population. In a previous study of U.S. blood donors, we reported that antibodies to unconjugated BPA-bound-to-albumin only occur in 13% of samples [15]. In our current study, we found a significant association between a subset of subjects that exhibited unconjugated BPA–HSA antibodies and also antibodies to alpha-synuclein.

Abnormal deposition of alpha-synuclein plays a central role in the pathogenesis of disorders such as PD and Dementia with Lewy bodies (DLB) [17]. Alpha-synuclein is a 140-amino acid protein that forms the major component of the abnormal filaments that comprise the Lewy bodies and Lewy neurites of PD and DLB. Alpha-synuclein is normally a natively unfolded intracellular protein that serves a role in synaptic transmission and augments transmitter release from the presynaptic vesicle [18]. The folding of alpha-synuclein into an insoluble aggregation in the extracellular regions can become neurotoxic—causing oxidative stress and vesicle trafficking—thereby promoting neurodegenerative diseases such as PD [19]. The aggregation of alpha-synuclein in the extracellular space can make it a susceptible target site for antibody complexes, and there is evidence that humoral immunity plays a role in the pathology of PD, which features elevated antibodies [20].

When intracellular alpha-synuclein misfolds and develops into extracellular aggregates, the humoral immune system can target them with protein-specific antibodies. These antibodies play a compensatory role during the neurodegenerative process to limit their neurotoxic effects [21]. Vaccinations with human alpha-synuclein in transgenic mice have led to the production of reactive antibodies that resulted in decreased accumulation of aggregated alpha-synuclein in neuronal cell bodies and reduced neurodegenerative changes [22]. It appears that antibodies against alpha-synuclein specifically target and aid in the clearance of extracellular alpha-synuclein protein by microglia, thereby preventing their neurotoxic actions on neighboring cells [23]. The use of exogenously developed alpha-synuclein antibodies also led to a reduced neuronal accumulation of alpha-synuclein aggregation [24]. The generation of these antibodies appears to serve as an early biomarker for the development of alpha-synuclein aggregation [16].

Evidence suggests that antibody production against specific assembled protein complexes can be used as a sensitive biomarker for neurodegenerative diseases [25,26]. In one study, autoantibodies toward major amyloidogenic proteins involved in PD Lewy bodies, including alpha-synuclein, were measured in the sera of early and late PD patients and controls using ELISA, Western blot, and Biacore surface plasmon resonance. Results demonstrated higher antibody levels toward alpha-synuclein in the sera of PD patients compared to controls. Yanamandra and colleagues reported a significant increase of 4- to 8-fold of the mean and median values of alpha-synuclein antibody levels in PD patients throughout the progression of their disease, compared to a very narrow distribution in healthy controls [26]. Papachroni and colleagues also detected high alpha-synuclein antibody tiers in 65% of patients with PD [25].

The identification of alpha-synuclein antibodies can serve as a potential biomarker to assess the development of alpha-synuclein aggregation years before the onset of disease in healthy subjects. In our study, we identified that those that have antibodies to BPA–HSA or PDI also have a considerable risk of having alpha-synuclein antibodies.

As with the limitation of an association study design, we cannot determine a direct causative relationship or the specific mechanisms involved between BPA–HSA antibodies or with PDI antibodies and the development of alpha-synuclein antibodies. Specific research must be conducted to understand the mechanisms of action of these relationships and whether healthy patients that exhibit these antibodies do in fact develop PD in the future.

With a clear understanding of the limitations of our study design and the restrictions of conclusions that can be extrapolated from our association data, we propose a theoretical model that is illustrated in Figure 4. In this model, bisphenol A crosses a breached blood-brain barrier and binds to the target enzyme PDI in the endoplasmic reticulum [27,28]. The production of antibodies to PDI then destroys PDI and contributes to protein misfolding and alpha-synuclein aggregation [29]. The development of these protein aggregates ultimately leads to the formation of alpha-synuclein antibodies [20]. This proposed model will require further research to adjudicate its hypothesis.

The role environmental chemicals play in the development of protein aggregates and neurodegenerative diseases by haptenation is a recent area of research. The interplay between chemical binding with proteins, the production of neoantigens, and immunological reactivity to neurological target sites provides some insights into how environmental chemicals may activate immune responses associated with neurodegeneration. Chemicals do not only induce toxic cellular reactions, but they also participate in binding to specific tissue proteins and generating immunological responses. These reactions can be identified with specific antibody levels. Further research is necessary to establish the causative role of these antibodies and their use in PD risk analysis.

## Figures and Tables

**Figure 1 toxics-07-00026-f001:**
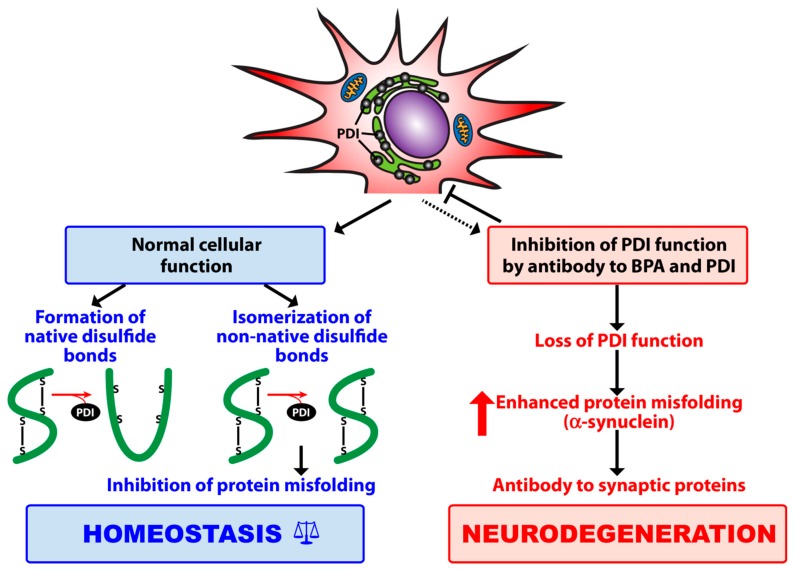
Contribution of protein disulfide isomerase (PDI) to protein folding, inhibition of protein misfolding, and maintenance of cellular homeostasis. The binding of bisphenol A (BPA) to its target enzyme (PDI) plus antibody to BPA and PDI results in loss of PDI functionality, which may contribute first, to alpha-synuclein aggregation and the misfolding of other synaptic proteins, and then to antibody production and neurodegeneration.

**Figure 2 toxics-07-00026-f002:**
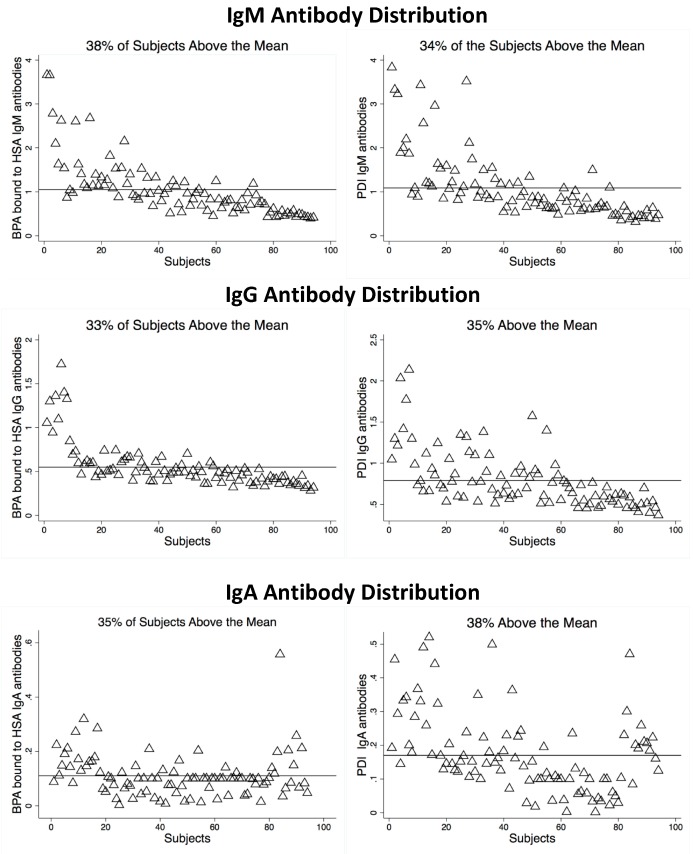
Optical density levels of BPA bound to HSA and PDI.

**Figure 3 toxics-07-00026-f003:**
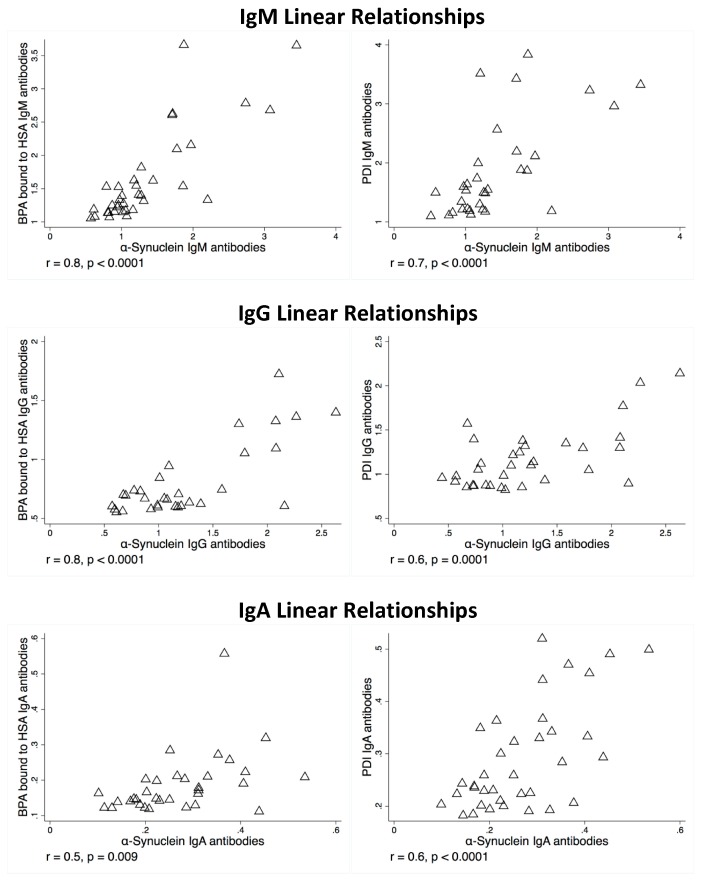
Linear relationship between alpha-synuclein antibodies and bisphenol A bound to HSA antibodies and alpha-synuclein antibodies and PDI antibodies.

**Figure 4 toxics-07-00026-f004:**
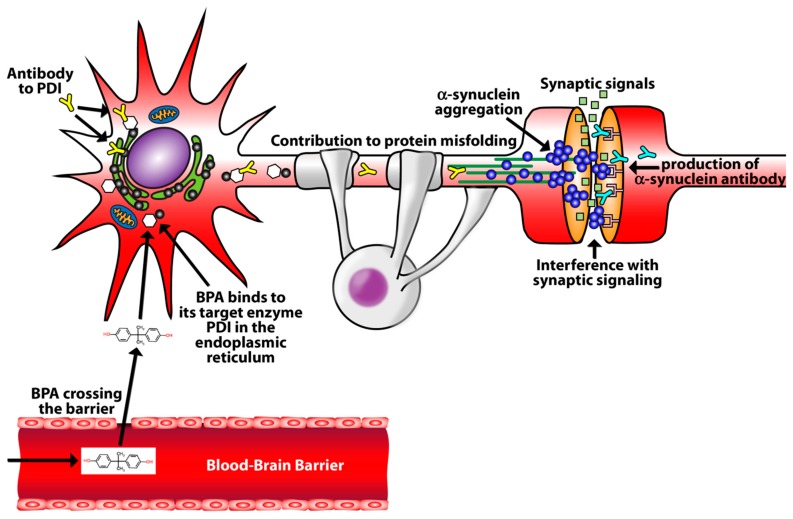
Theoretical mechanism of BPA promotion of alpha-synuclein aggregation and the development of alpha-synuclein antibodies. Bisphenol A crosses a breached blood-brain barrier and bind to the target enzyme PDI in the endoplasmic reticulum. The development of antibodies to PDI contributes to protein misfolding and alpha-synuclein aggregation. Antibodies form against alpha-synuclein.

**Table 1 toxics-07-00026-t001:** Risk ratios for developing elevated alpha-synuclein antibodies.

Elevated Exposure Antibody	Risk Ratio	Confidence Interval	*p*-Value
BPA–HSA IgM	6.7	3.5–13.1	<0.0001
PDI IgM	13.6	5.2–35.3	<0.0001
BPA–HSA IgG	4.8	2.4–9.6	<0.0001
PDI IgG	3.4	1.9–6.1	<0.0001
BPA–HSA IgA	9.8	3.8–25.6	<0.0001
PDI IgA	10.8	4.1–18.0	<0.0001

BPA–HSA = Bisphenol A bound to serum albumin antibodies, PDI = Protein disulfide isomerase. Elevated antibodies refer to antibodies above the mean. There was no significant risk for alpha-synuclein antibodies when the levels of BPA–HSA or PDI antibodies were below the mean.

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
