# Peer review of "The Associations between Immunological Reactivity to the Haptenation of Unconjugated Bisphenol A to Albumin and Protein Disulfide Isomerase with Alpha-Synuclein Antibodies"

_toxics, 2019, doi:10.3390/toxics7020026_

Round 1

Reviewer 1 Report

Thank you very much for giving me the opportunity to review the article entitled: "The Associations between Immunological Reactivity to the Haptenation of Unconjugated Bisphenol-A to Albumin and Protein Disulfide Isomerase with Alpha-Synuclein Antibodies”. The topic is interesting and has great potential if explored in more detail, however, I have some issues that prevent me from accepting this paper in its present form.

Major:

1-In the methods part, you claim that “Tests samples were selected and arranged to reflect the adult population of the United States by sex and race. Adult subjects were separated by gender and then by racial distribution to reflect the U.S. population. Serum samples included: 60% Caucasians; 20% Hispanics; and 20% African American and other races”. However, data are shown as a whole, and results were not analyzed separately by gender or race. Please discuss the rationale of doing it this way.

2-Were blood samples from healthy donors, or is it possible to correct for cardiovascular, metabolic or autoimmune disease? Please explain, you only claim that they were “hepatitis C, RNA, hepatitis B surface antigen, antibodies to HIV, HIV‐1 RNA, and syphilis” negative.

3-One improvement to the study would be to evaluate BPA concentration in the blood samples and correlate with alpha‐synuclein antibodies. If possible, performing this analysis is highly recommended.

Author Response

Thank you for your comments on our manuscript.  We would like to address your concerns.

1 – When analyzing the data by gender, race, or age we did not find any significant outcomes to report. We have added the following sentence to section 2.5 Statistical analysis:

“We did not have adequate power to stratify our samples for gender, sex, or race in our study. Our analyses were only powered to analyze groups above and below the mean.”

2 - Our samples were purchased from Innovative Research Inc.  (Southfield, MI, USA) and defined as healthy donors.  We did not have the resources to evaluate for the possibility of other cardiovascular, metabolic, or autoimmune diseases in this initial study.

We have added the following sentence to section 2.1 Blood Samples:

“There was no assessment for cardiovascular, metabolic, or autoimmune diseases and we did not have any further medical information of the subjects we used in our samples.”

3 –BPA concentrations are ubiquitously elevated in the majority of the population and therefore we did not include it in our study design. The aim of our study was to determine if immunological reactivity to BPA-HSA had any associations to alpha-synuclein antibodies. We decided that these reactions occur on only a limited subset of the population and therefore we did not include BPA concentration in our study design since our focus was limited to investigating associations with chemical-immune reactivity instead of BPA concentrations.

We have modified the third paragraph of section 4.0 Discussion to:

“We did not measure concentrations of BPA in our study since elevated BPA levels in biofluids is a ubiquitous finding. The perpetual exposure of BPA to humans is an established feature of modern society due to the extensive use of plastic products. Detectible levels of BPA in human serum and urine samples are identified in the majority of the population. More than 90% of the samples tested in the United States (U.S.) population have measurable BPA levels in biofluids (4). However, antibodies produced against BPA-bound-to-protein are an independent feature of only a subset of the population. In a previous study of U.S. blood donors, we reported that antibodies to unconjugated BPA-bound-to-albumin only occur in 13% of samples [15]. In our current study, we found a significant association between a subset of subjects that exhibited unconjugated BPA-HSA antibodies and also antibodies to alpha-synuclein.”

We are grateful for your comments and for your assistance in helping us improve our manuscript.

Reviewer 2 Report

This study showed associations between unconjugated BPAbound to human serum albumin (BPAHSA) antibodies and alphasynuclein antibodies and between PDI antibodies and alphasynuclein antibodies using EIA methods.  The results were very unique and interesting biological activities of BPA .  Authors mentioned as a conclusion that there are significant associations and risk between unconjugated BPAHSA and PDI antibodies for alphasynuclein antibodies.

 However, it is not clear that what is definition of “risk” in this study. 

Is the concentration of BPA used here too high? Authors should also explain this point.

Author Response

Thank you for your comments on our manuscript.  We would like to address your concerns.

In our study we evaluated risk ratios for elevated alpha-synuclein antibodies (IgG, IgM, and IgA levels above the mean) of samples that presented with elevated BPA-HSA antibodies or elevated PDI antibodies compared to those that did not have these antibody elevations. We did not measure concentrations of BPA in the serum.

We have revised the conclusion section of the Abstract to the following sentence:

"We conclude that there are significant associations and risk between unconjugated BPA-HSA antibodies and PDI antibodies for developing alpha-synuclein antibodies."

We have also revised Section 3. Results in our manuscript to the following paragraph to make our risk analysis more clear:

"Risk ratio (RR) calculations were conducted in two groups that included; (1) subjects with antibodies above the mean and (2) subjects with antibodies below the mean, for BPA-HSA antibodies and PDI antibodies to determine if there were any risk for exhibiting elevated alpha-synuclein antibodies.  There was no identified risk for exhibiting elevated alpha-synuclein antibodies with subjects that had their BPA-HSA antibodies or PDI antibody levels below the mean. However, there was a significant risk (RR between 3.4-16.7) of exhibiting alpha-synuclein antibodies if the subjects had elevated (above the mean) BPA-HSA antibodies or PDI antibodies. These risk ratios are listed in table 1."

We are grateful for your comments and for your assistance in helping us improve our manuscript.

Reviewer 3 Report

It is a very novel manuscript and I would like to recommend for publication.

It would be more interesting to stratify the study with patients' gender and age 

The results would be more valid if the antibody titer rather than OD values are presented as the actual concentration would be more comparable to each other than OD value alone.

Author Response

Thank you for your comments on our manuscript.  We would like to address your concerns.

1 – When analyzing the data by gender, age, or race we did not find any significant outcomes to report. We have added the following sentence to section 2.5 Statistical analysis:

“We did not have adequate power to stratify our samples for gender, sex, or race in our study. Our analyses were only powered to analyze groups above and below the mean.”

2- To address the OD values versus titer comment, because the indices and ODs were equal, there was no need to report the titer. We have added the following paragraph to section 2.4 Measurement of Antibody Enzyme-Linked Immunosorbent Assay (ELISA):

Additionally, in each assay, we included sera of patients with Parkinson’s disease with moderate to high levels of alpha-synuclein, PDI, and BPA antibodies. An optimal dilution of the sera at 1:200 -1:400 and incubation for 45 minutes resulted in optical densities of 1.1 +/- 5%, from which we chose the three best to use as a calibrator in each assay, one for BPA-HSA, one for PDI, and one for alpha-synuclein. Then, the ODs were converted to ELISA index using the following formula:

ELISA Index =

OD of sample – OD of   blank

OD of calibrator – OD   of blank

Because the OD of our calibrators was 1.1 and the blank wells were about 0.1 +/- 10%, the calculations of our indices were 1.0 +/- 5% which were very close to ODs, and therefore we decided to report the original ODs in our analyses

Thank you for your comment and for your assistance in helping improve our manuscript.

Round 2

Reviewer 1 Report

My concerns were adequately addressed. The paper is suitable for publication.